


# Characteristics of supersaturation in mid-latitude cirrus clouds and their adjacent cloud-free air

Georgios Dekoutsidis[1], Silke Groß[1], Martin Wirth[1], Martina Krämer[2,3], Christian Rolf[2]

[1]Institut für Physik der Atmosphäre, Deutsches Zentrum für Luft- und Raumfahrt (DLR), Oberpfaffenhofen, 82234 Wessling, Germany
[2]Institute for Energy and Climate Research (IEK-7), Research Center Jülich, 52425 Jülich, Germany
[3]Institute for Atmospheric Physics (IPA), Johannes Gutenberg University, Mainz, Germany

*Correspondence to*: Georgios Dekoutsidis (Georgios.dekoutsidis@dlr.de)

**Abstract.** Water vapor measurements of mid-latitude cirrus clouds, obtained by the WALES (WAter vapor Lidar Experiment in Space) lidar system during the Mid-Latitude Cirrus (ML-Cirrus) airborne campaign, that took place in spring of 2014 over central Europe and the NE Atlantic Ocean, are combined with model temperatures from the European Centre for Medium-Range Weather Forecasts (ECMWF) and analysed. Our main focus is to derive the distribution and temporal evolution of humidity with respect to ice within cirrus clouds and in their adjacent cloud-free air. We find that 34.1 % of in-cloud data points are supersaturated with respect to ice. Supersaturation is also detected in 6.8 % of the cloud-free data points. When the probability density of the Relative Humidity over ice (RHi) is calculated with respect to temperature for the in-cloud data points from the ML-Cirrus data-set there are two peaks. One around 225 K and close to saturation, RHi = 100 % and a second one at colder temperatures around 215 K in subsaturation, RHi = 90 %. These two regions seem to represent two cirrus cloud categories: in-situ formed and liquid-origin. Regarding their vertical structure, most clouds have higher supersaturations close to the cloud-top and become subsaturated near the cloud bottom. Finally, we find that the vertical structure of RHi within the clouds is also indicative of their life stage. RHi skewness tends to go from positive to negative values as the cloud ages. RHi modes are close to saturation in young clouds, supersaturated in mature clouds and subsaturated in dissipating clouds.

## 1 Introduction

Cirrus clouds greatly affect the Earth's climate system, primarily through their impact on the radiation budget (Manabe and Strickler, 1964; Cox, 1971; Ramanathan et al., 1983; Liou, 1986; Stephens et al., 1990; Chen et al., 2000; Gasparini et al., 2018). Despite that, the exact effects are still not certain (IPCC, 2007, 2013). Typically, they are considered to have a net warming effect (Chen et al., 2000), but this can differ between clouds with varying characteristics (Joos et al., 2014; Krämer et al., 2020). The radiative effects of mid-latitude cirrus clouds strongly depend on their microphysical properties, such as ice crystal shape, size and concentration (Stephens et al., 1990; Haag and Kärcher, 2004; Fusina et al., 2007), which are in turn largely determined by their initial freezing mechanism and ambient conditions (e.g. temperature, available aerosol,





turbulence) (Heymsfield, 1977; Khvorostyanov and Sassen, 1998; Kärcher and Lohmann, 2003; Seifert et al., 2004; Gensch et al., 2008; Krämer et al., 2016; Luebke et al., 2016). Modelling of these small-scale microphysical processes —for which much still remains unknown— proves to be a challenge (Burkhardt, 2012; Kärcher, 2017). Thus, most models use parametrizations and approximations, leading to inaccuracies in the prediction of cirrus clouds, their radiative effects and

interactions with aerosols (Kärcher et al., 2006; Liu and Penner, 2005; Phillips et al., 2008). To improve the model predictions and be able to determine with greater precision the effects that mid-latitude cirrus clouds have on the climate, a better understanding of the micro- macrophysical and optical properties of these clouds is needed (Haag et al., 2003; Di Girolamo et al., 2009; Groß et al., 2014).

Measurements of cirrus clouds can be conducted either in-situ or with remote sensing instruments, such as LIDAR. For

in-situ measurements the instruments are aircraft-bound and flown through the cirrus clouds. With this method specific clouds can be selected and parts of the cloud in different evolutionary stages can be measured. A downside is that measurements can be conducted only at one selected flight level at a time. Thus, in-situ measurements do not provide information about the vertical structure of the clouds. Lidar measurements on the other hand have the advantage of providing a 2D curtain of the measured clouds, giving information also on their vertical structure. Lidar instruments can be ground-

based, airborne or satellite-bound. Ground-based lidar are capable of measuring cirrus clouds with great spatial and temporal resolution (Comstock et al., 2002, 2004), but they lack versatility. Ground-based lidar can only measure clouds that happen to pass over them at one location. This way the collected data can be used for statistical analysis, rather than case studies, as the measured clouds cannot be chosen and it is uncertain which parts of the clouds and at which evolutionary stages will overpass the lidar. Satellite remote sensing with lidar also provides 2D measurements of cirrus clouds, but they lack

accuracy, compared to the other methods, because of the distance between instrument and cloud and can only measure clouds along their flight tracks. Finally, airborne lidar measurements are the most versatile and accurate. Specific clouds can be selected and the measurements can be conducted in such a way that many different parts and evolutionary stages of the cloud are measured. That way, the vertical structure can be measured for different stages of the cloud's life-cycle.

Various lidar measurement techniques have been developed with a multitude of applications, capable of measuring

polarization, temperature, windspeed, backscatter and others. The High Spectral Resolution Lidar (HSRL) technique, filters out the molecular backscatter and provides the aerosol backscatter, with which clouds are identified. Differential Absorption Lidar (DIAL) systems use laser frequencies on and off absorption lines of gases, such as water vapor in order to determine their mixing ratio with great vertical and temporal resolution, especially when airborne (Ehret et al., 1993; Bösenberg, 1998; Browell et al., 1998; Chen et al., 2002; Groß et al., 2014).

Both in-situ and airborne lidar measurements face the problem of needing to reach very high altitudes. Few research aircraft are capable of reaching these altitudes to perform measurements (Groß et al., 2014). The German research aircraft HALO (High Altitude and LOng range) (Krautstrunk and Giez, 2012) provides a platform that is capable of reaching the altitudes needed in order to successfully measure cirrus clouds (Groß et al., 2014). During the ML-Cirrus campaign, which took place over central Europe and the Northeast Atlantic in spring of 2014 (Voigt et al., 2017), HALO was equipped with



the Differential Absorption (DIAL) and High Spectral Resolution (HSRL) lidar system, WALES (Wirth et al., 2009). By combining both techniques, WALES provides high temporal and spatial resolution curtains of the optical properties of clouds and collocated water vapor measurements. Here we combine the water vapor measurements taken by WALES during the ML-Cirrus campaign with model temperatures from ECMWF and calculate the relative humidity over ice (RHi) for these clouds.

Our goals are to study the distribution of humidity with respect to ice within and in the vicinity of mid-latitude cirrus clouds, determine the vertical structure of RHi within them and its temporal evolution and also to investigate the differences between clouds with different formation processes. The questions we want to answer are: Is supersaturation detected in cirrus clouds and the cloud free air at the mid-latitudes? How is it distributed and what values does it reach? What is the vertical structure of RHi within these clouds? What cloud regions can be determined by the vertical distribution? How does

the formation process affect the relative humidity? How does the relative humidity change during the life cycle of a cloud?

## 2 Data & Method

### 2.1 ML-Cirrus Campaign

In March and April of 2014, the ML-Cirrus campaign took place. Main objective of this campaign was to study the

microphysical, macrophysical, radiative and optical properties of natural and anthropogenic cirrus clouds over the mid-latitudes. For this, the High Altitude and Long-Range Research aircraft HALO was used. It was equipped with in-situ as well as remote sensing instruments and

performed research flights over Central Europe and the NE Atlantic. More details about this campaign can be found in Voigt et al. (2017).

### 2.2 WALES System

One of the instruments onboard the HALO aircraft was the

WALES lidar system. WALES is a high-spectral-resolution (HSRL) and differential absorption (DIAL) lidar (Esselborn et al., 2008; Wirth et al., 2009). It provides 2D measurements of extinction coefficient, backscatter coefficient, aerosol depolarization and water vapor mixing ratio. In this study we use data taken by the WALES system during ten missions of the ML-Cirrus campaign (Figure 1). We focus on the water vapor mixing ratio measured with the DIAL technique. These measurements have been found to be accurate and

suitable for the study of ice clouds: Groß et al. (2014), used water vapor measurements from WALES and found a good

**Figure 1: Lidar Legs of the research flights conducted during the ML-Cirrus campaign. The colour scale represents the mean RHi of the vertical column from cloud base to cloud top. The white colour represents columns for which the mean RHi is 70 % or less, or regions where no ice clouds where measured along the flight track.**



agreement with in-situ measurements. Kiemle et al. (2008) estimated the statistical error of the water vapor retrieval to be about 5 %, although the exact value is dependent on various parameters that differ for individual measurements. Errors that arise due to the high spatial inhomogeneity of the backscatter within cirrus clouds are kept below 5 % by filtering. Finally, the Rayleigh-Doppler effect is corrected in the retrieval algorithm, leaving an error of less than 2 % (Groß et al., 2014). In

depth description, technical characteristics and accuracy of WALES can be found in the abovementioned references.

## 2.3 Cirrus Classification

A classification of cirrus clouds, based on their formation mechanism and microphysical properties was presented by Krämer et al. (2016) and also Luebke et al. (2016). They distinguish between two types of cirrus. The first type is in-situ origin cirrus, which form directly as ice. The second type is liquid-origin cirrus. They originate from mixed phase clouds, whose

liquid droplets freeze latest when they reach cold enough temperatures (< 235 K). In our study we use the same classification and group our clouds based on 24-hour backwards trajectories. They are calculated with wind data from the ECMWF reanalysis dataset ERA – Interim (ECMWF, 2011). For vertical transport, diabatic heating rates are used with the trajectory module of the Chemical Lagrangian Model of the Stratosphere (CLaMS) (McKenna et al., 2002). For some lidar legs in-situ as well as liquid-origin clouds where measured during one flight. Furthermore, some clouds had an in-situ layer -most

commonly the top layer- and a liquid-origin part. In these cases, the data set was split and grouped accordingly.

## 2.4 Methodology

The Relative Humidity over ice (RHi) largely controls the formation and nucleation mechanism of cirrus clouds which in turn has a great effect on their characteristics (Heymsfield and Miloshevich, 1995; Comstock et al., 2002; Haag et al., 2003; Krämer et al., 2009; Sakai et al., 2014). Based on that we choose RHi as our main parameter in this study. We calculate RHi

from the retrieved water vapor mixing ratio and model temperature from ECMWF (Esselborn et al., 2008). We use equations from Huang, (2018) for the calculation of the saturation vapor pressures of water and ice. A similar method has also been applied by Groß et al. (2014) and Urbanek et al. (2018). Groß et al. (2014) compared the model temperature with in-situ measurements and found that ECMWF temperature data induces an error of about 10 to 15 % in the calculated RHi. Despite that, they concluded that the ECMWF model temperature is suitable for the study of cirrus clouds at the mid-latitudes.

In the next step we construct and apply a cloud mask, in order to separate the cirrus clouds and the cloud-free air around them. We set thresholds for the backscatter ratio (BSR), particle linear depolarization ratio (PLDR), temperature and altitude of each data point (Table 1). Groß et al. (2014) and Urbanek et al. (2018) find that a wide range of BSR values can be used as a threshold between in-cloud and cloud-free data points. They chose a BSR threshold of 2 and 4, respectively. From our own analysis we find the optimal BSR to be 3. The PLDR threshold is selected at 20 %, as this is indicative of ice crystals

(Gobbi, 1998; Chen et al., 2002; Comstock et al., 2004). The temperature of 235 K is the threshold below which in-situ cirrus clouds form (Ström et al., 2003; Kärcher and Seifert, 2016; Krämer et al., 2016; Gasparini et al., 2018; Urbanek et al.,



2018). The cloud-free data in the vicinity of the cirrus clouds, are the data points from the whole lidar leg curtain after the detected clouds are removed. We choose to only keep data points with an altitude higher than 7 km as we only want areas where ice crystals would most probably be able to form (Ström et al., 2003; Comstock et al., 2004; Gasparini et al., 2018; Kärcher, 2017). We consider the vicinity around cirrus clouds as a maximum horizontal distance of 250 km from the cloud edges and altitudes from 7 km to 12 km as we mostly detect cirrus clouds in this range.

**Table 1: Thresholds of the cloud-mask.**

|  | In-Cloud | Cloud-free |
|---|---|---|
| BSR | $\geq 3$ | $< 3$ |
| PLDR | $\geq 20\ \%$ | $< 20\ \%$ |
| Temperature | $< 235$ K | - |
| Height | - | $> 7$ km |

Finally, as a means to get some insight on the microphysics of the clouds and the ice nucleation processes, we calculate thresholds for heterogeneous nucleation (HET), homogeneous nucleation (HOM) and water saturation (Urbanek et al., 2017, their Table 1, and original formulations from Krämer et al., 2016). The water saturation threshold is the limit above which water droplets can form in addition to ice crystals. In-situ HOM nucleation occurs when supercooled solution droplets (SSP) are lifted up to altitudes with very low temperatures (<235 K). For HET nucleation to take place, ice nucleating particles (INP) are needed. Different INP have different freezing thresholds. We specify a high threshold and a low threshold. For the high threshold we consider INP which are inefficient for ice formation such as coated soot and for the low threshold we consider the INP to be mineral dust which is more efficient as an INP and more abundant (Pruppacher and Klett, 1997; Kärcher and Lohmann, 2003; Gensch et al., 2008; Hoose and Möhler, 2012; Cziczo et al., 2013; Krämer et al., 2016; Ansmann et al., 2019).

## 3 Results

### 3.1 Overview

In Table 2, we present the most common values of the RHi distributions (RHi modes) and supersaturation percentages inside the cirrus clouds and in the cloud-free air around them for the whole data set, the in-situ and the liquid-origin clouds. In order to get a more detailed insight in the supersaturation we define three bins of RHi, 100 %–120 %, 120 %–140 % and > 140 %. RHi 120 % and 140 % can be considered approximate thresholds for HET and HOM respectively (Koop et al., 2000; Haag et al., 2003; Comstock et al., 2004; Khvorostyanov and Curry, 2009; Kärcher, 2012).

The frequency distribution of RHi for the in-cloud data points of the whole data set has a mode value of 96 %. Around 34 % of the in-cloud data points are supersaturated with respect to ice. 30.3 % have RHi up to 120 %, 3.3 % have RHi between 120 % and 140 % and 0.5 % have RHi higher than 140 %. For the cloud-free data points the frequency distribution





of RHi has a mode value of 68 %. 6.8 % of the cloud-free data points are supersaturated. 5.7 % have RHi between 100 %

and 120 %, 1 % is in the range 120 % to 140 % and 0.2 % have RHi higher than 140 %.

**Table 2: Results from the analysis of the probability density functions (PDF) of RHi. RHi mode: most common value of dataset.**

|  | In cloud | | | Cloud-free air adjacent to | | |
|---|---|---|---|---|---|---|
|  | All clouds | In-situ | Liquid-origin | All clouds | In-situ | Liquid-origin |
| # of data points | 3,204,381 | 1,110,675 | 2,093,857 | 2,762,080 | 948,638 | 1,813,300 |
| RHi mode [%] | 96 | 96 | 96 | 68 | 56 | 84 |
| RHi PDF [%] |  |  |  |  |  |  |
| ≥ 100 % | 34.1 | 30.8 | 36 | 6.8 | 5.7 | 7.5 |
| RHi PDF [%] |  |  |  |  |  |  |
| 100 % - 120 % | 30.3 | 26.8 | 32.3 | 5.7 | 4.5 | 6.3 |
| 120 % - 140 % | 3.3 | 3.4 | 3.3 | 1 | 1.1 | 1 |
| ≥ 140 % | 0.5 | 0.6 | 0.5 | 0.2 | 0.2 | 0.2 |

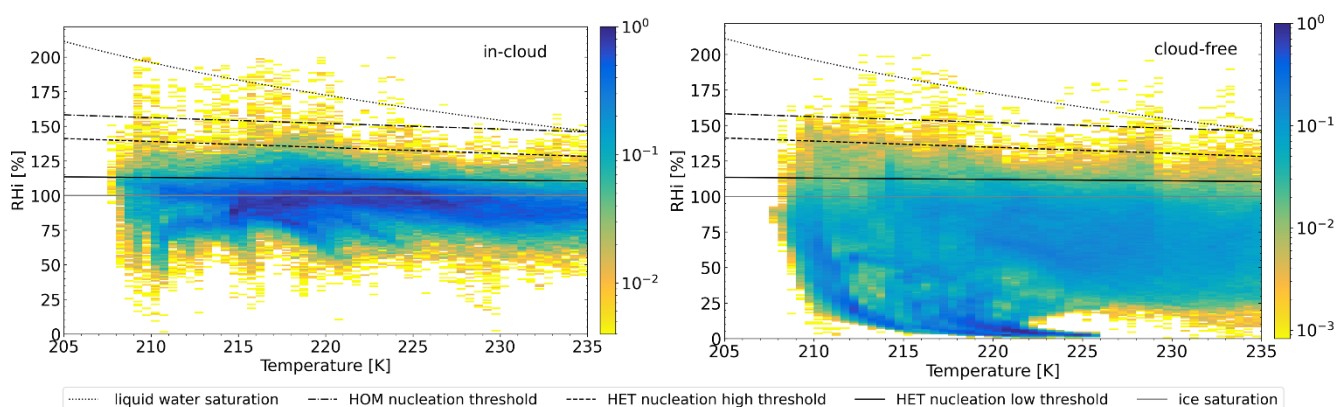

**Figure 2: Probability densities of the Relative Humidity over Ice (RHi) with respect to ambient temperature. Left: Calculated for the in-cloud data points. Right: Calculated for the cloud-free data points. The bin sizes are 0.5 K and 1 % RHi. The gray line depicts the ice saturation threshold at RHi = 100 %. The solid black line is the low threshold for heterogeneous nucleation (HET). The ice nucleating particle (INP) for this threshold corresponds to mineral dust, which is efficient as an initiator of ice formation. The dashed line represents the high threshold for HET nucleation. Here the INP corresponds to coated soot which is not activated**
**as easy as mineral dust. The dot-dashed line indicates the threshold at which homogeneous nucleation (HOM) can take place. During HOM, ice crystals form without the need of INP. The dotted line is the supercooled liquid water saturation line.**

Aiming to gain a more detailed view into the characteristics of RHi and get an insight on the nucleation mechanisms we plot

the probability density of RHi with respect to the ambient temperature, accompanied by the ice and water saturation (Huang,

2018) lines, low/high HET (Gensch et al., 2008 as shown in Figure 4 of Krämer et al., 2016) and the HOM (Koop et al.,

2000 as shown in Figure 4 of Krämer et al., 2016) regime (Figure 2). The in-cloud data points reach temperatures down to





207 K and are most frequently detected close to ice saturation (RHi = 100 %) for the entire temperature range. Their

distribution is bimodal: One peak can be seen between 210 K and 225 K and close to ice saturation, and a second one around

215 K and below saturation at RHi around 90 %. An increase in supersaturated points can be seen for temperatures between

215 K and 225 K. Regarding the nucleation processes, most of the supersaturated data points are in the HET regime, with the

highest probabilities close to the low HET threshold. The probability density drops significantly above the high HET

threshold. Very few data points are found above the HOM threshold. Finally, there is also a very small probability of

detection over the water saturation threshold, which probably stems from the uncertainties in the water vapor measurements,

the model temperatures or both.

The majority of the cloud-free data points are detected below the lower threshold for HET nucleation. Some data points

reach up to the high HET threshold. The probability for detection up to the high HET threshold is higher for colder

temperatures. The probability of data points exceeding the high HET and HOM thresholds is very low. A peak of the

probability density can be seen around 220 K and relative humidities close to zero, forming a coma shape. This feature is

detected in many cases and is indicative of air masses with various temperatures and a constant water vapor mixing ratio

around 1.5 ppmv, which is the minimum value observed in

the upper troposphere (Krämer et al., 2009) or data points

measured in the stratosphere.

In Figure 3, we present the vertical distribution of RHi

within the mid-latitude cirrus clouds in our data set. For

each cloud the in-cloud data points are divided in four

groups based on their RHi value. The four groups

represent the subsaturated, low HET, high HET and HOM

regimes. The cloud-base and cloud-top are determined and

the relative location of every data point with respect to

cloud top is calculated in bins of 10 % distance between

top and base. From the cloud top up to 20 % cloud depth

high supersaturations with RHi values above 140 % are

most frequent. From the 20th to the 40th percentile, the

dominating group is supersaturations between 120 % and

140 %. From the 40th to the 60th percentile (midway

200 between cloud-top and bottom) supersaturations of up to

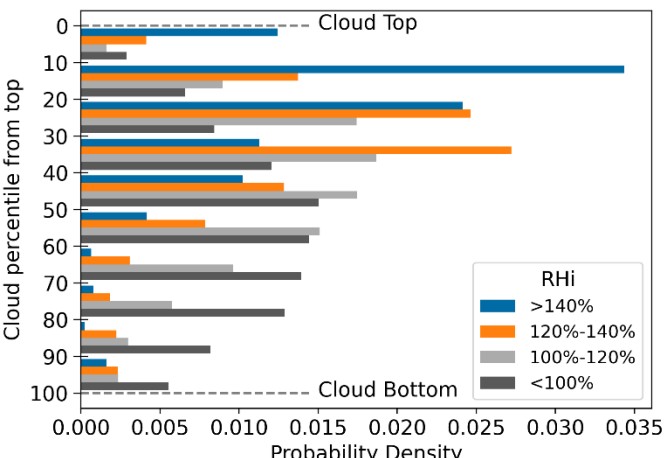

**Figure 3: Probability densities of RHi in relative location to cloud top. RHi values of in-cloud data points are grouped into four groups approximately representing subsaturation (dark grey), low HET regime (grey), high HET regime (orange) and HOM regime (blue). For every group the probability density in bins with respect to relative location to cloud top is plotted. Example: the bin from the 20th to the 30th percentile contains the probability of occurrence of the four RHi groups in a slice of the cloud located 20 % to 30 % of the distance between cloud top and cloud base.**

120 % are most frequent. From 60 % cloud depth and until the cloud bottom mostly subsaturated data points are detected. In

summary, high supersaturations are detected at cloud top and lower supersaturations gradually become dominant towards the

middle of the clouds. From around the mid-point of the clouds and until the cloud bottom most data points are subsaturated.





For around 20 % of the cases the uppermost layer has lower supersaturation than deeper layers or is even subsaturated (not
shown).

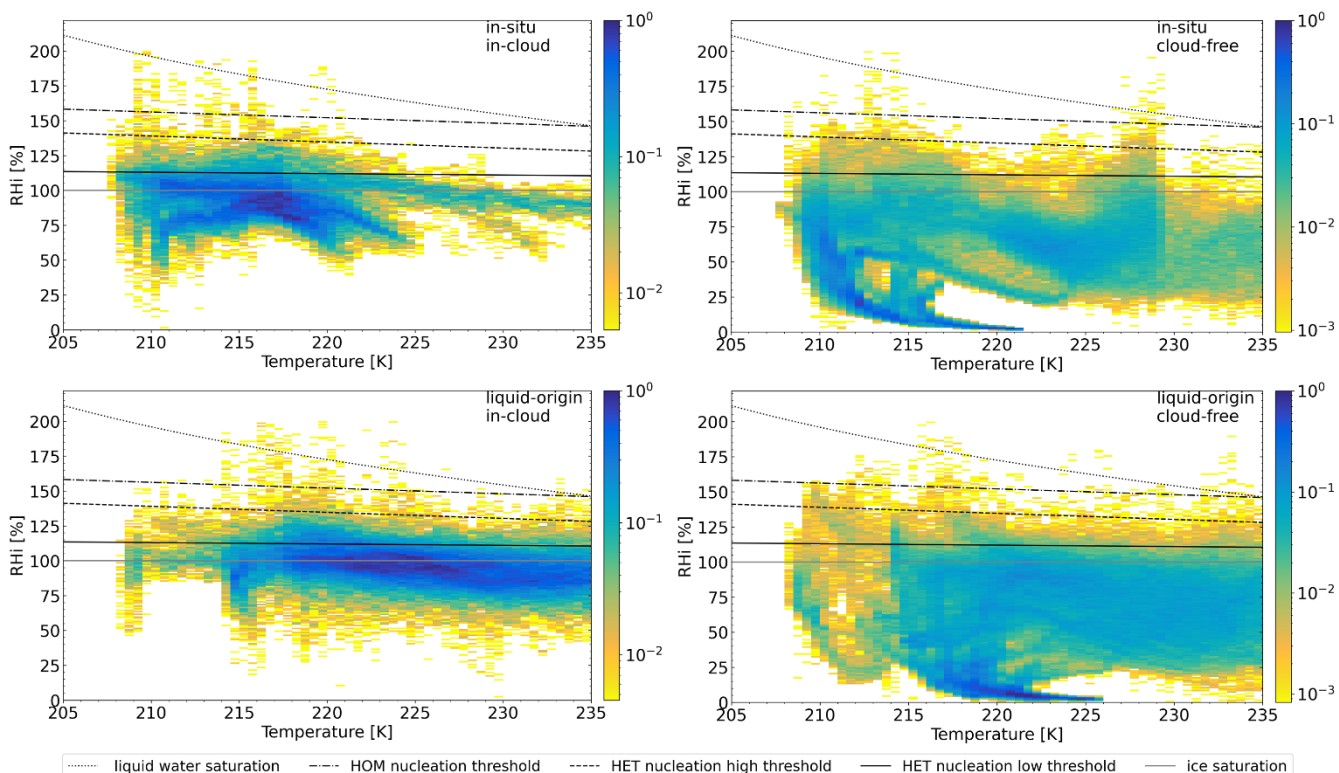

**Figure 4: PDFs of RHi with respect to ambient temperature. Top Left: in-situ in-cloud data points. Top Right: in-situ cloud-free data points. Bottom Left: liquid-origin in-cloud data points. Bottom Right: liquid-origin cloud-free data points. In detail explanation see Figure 2.**

## 3.2 Comparison of in-situ and liquid-origin clouds

The frequency distribution of RHi for the in-situ formed cirrus clouds has a 96 % RHi mode value (Table 1). Around 31 % of the data points are supersaturated with respect to ice. 26.8 % have low supersaturation, RHi up to 120 %, 3.4 % have RHi values between 120 % and 140 % and 0.6 % have high supersaturations with RHi being above 140 %. The frequency distribution for the liquid-origin clouds also has a mode value of 96 %. Supersaturation has an occurrence frequency of 36

%, which is higher than for the in-situ clouds. 32.3 % of the in-cloud data points have low supersaturations, RHi < 120 %, 3.3 % have RHi values between 120 % and 140 % and 0.5 % have higher supersaturations with RHi exceeding 140 %.

The ambient conditions at which in-situ and liquid-origin clouds form are different. This leads to differences in the distribution of RHi inside the clouds as well as in their surrounding cloud-free air. Although cloud-free air cannot be classified as in-situ or liquid-origin we use the terms in-situ and liquid-origin cloud-free for simplicity to describe the air

adjacent to in-situ and liquid- origin clouds respectively. For the cloud-free data points in the vicinity of in-situ clouds the mode of the frequency distribution of RHi is 56 %. Supersaturation occurs for 5.7 % of the data points. 4.5 %, have RHi





between 100 % and 120 %, 1.1 % have RHi values from 120 % to 140 % and finally 0.2 % have high supersaturations with RHi higher than 140 %. The frequency distribution of RHi for cloud-free data points around liquid-origin clouds has a higher mode at 84 %. Supersaturation has an occurrence of 7.5 %. 6.3 % in the low supersaturation regime, RHi up to 120 %, 1 %

between for RHi between 120 % and 140 % and 0.2 % higher than 140 %.

In Figure 4, we plot the in-cloud and cloud-free frequency distributions of RHi with respect to ambient temperature for both cloud groups. In-situ cirrus clouds are more frequently measured at temperatures below 225 K. The peak is located at a temperature around 217 K and an RHi of 90 %. Most data points are below the low HET threshold, but can also reach the high threshold. The liquid-origin clouds are mostly measured in warmer environments, above 215 K. Most data points are

close to saturation. The peak can be located around a temperature of 225 K and an RHi of 100 %. The probability density is significant up to the high HET threshold, mostly at around 220 K. The two peaks detected in Figure 2, seem to correspond to the two cloud types.

The cloud-free data points around in-situ clouds are mostly subsaturated and reach temperature as low as 208 K. The few data points with an RHi above 100 % are detected for temperatures below 220 K, reaching up almost to the high HET

threshold, although a peak to higher RHi is also located at around 228 K. The cloud-free data points in the vicinity of liquid-origin cirrus clouds are mostly measured in warmer temperatures, above 215 K. Low supersaturations, below the low HET threshold are detected on a wide temperature range, 214 K-235 K. The coma shape seen in Figure 2 is also present in the cloud-free distributions of both cloud groups.

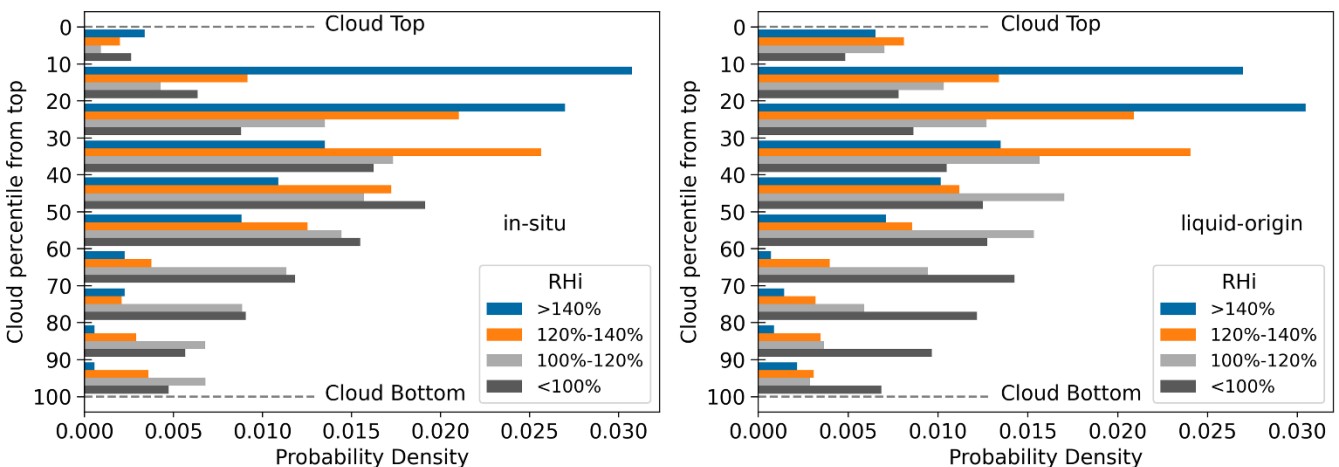

**Figure 5: Probability densities of RHi in relative location to cloud top. Left: in-situ clouds. Right: Liquid-origin clouds. For in-detail description see Figure 3.**

In Figure 5 we present the vertical distribution of RHi for in-situ and liquid-origin cirrus clouds, following the same method as in Figure 3. Both cloud types seem to have a similar distribution of RHi as the one described in Section 3a. High supersaturations around cloud top getting lower towards cloud middle and then subsaturated until cloud base. Despite

following this general form each group has individual characteristics. For the in-situ clouds the cloud base is dominated by





data points with low supersaturations < 120 % rather than subsaturated points. Subsaturated data points become the dominant group below cloud middle and down to the 80th percentile of cloud depth. Above the cloud middle the distribution follows the general form. Liquid-origin clouds follow the general vertical distribution from cloud base and up to the 10th percentile. The cloud top —down to the 10th percentile of cloud depth— is most frequently populated by data points with RHi values 250   lower than the deeper parts of the cloud.

## 3.3 Case Studies of Temporal Evolution

We choose two special cases, so as to study the temporal evolution of cirrus clouds and more specifically the changes in RHi through various stages of the cloud's lifetime.

### 3.3.1 Mesoscale Convective System

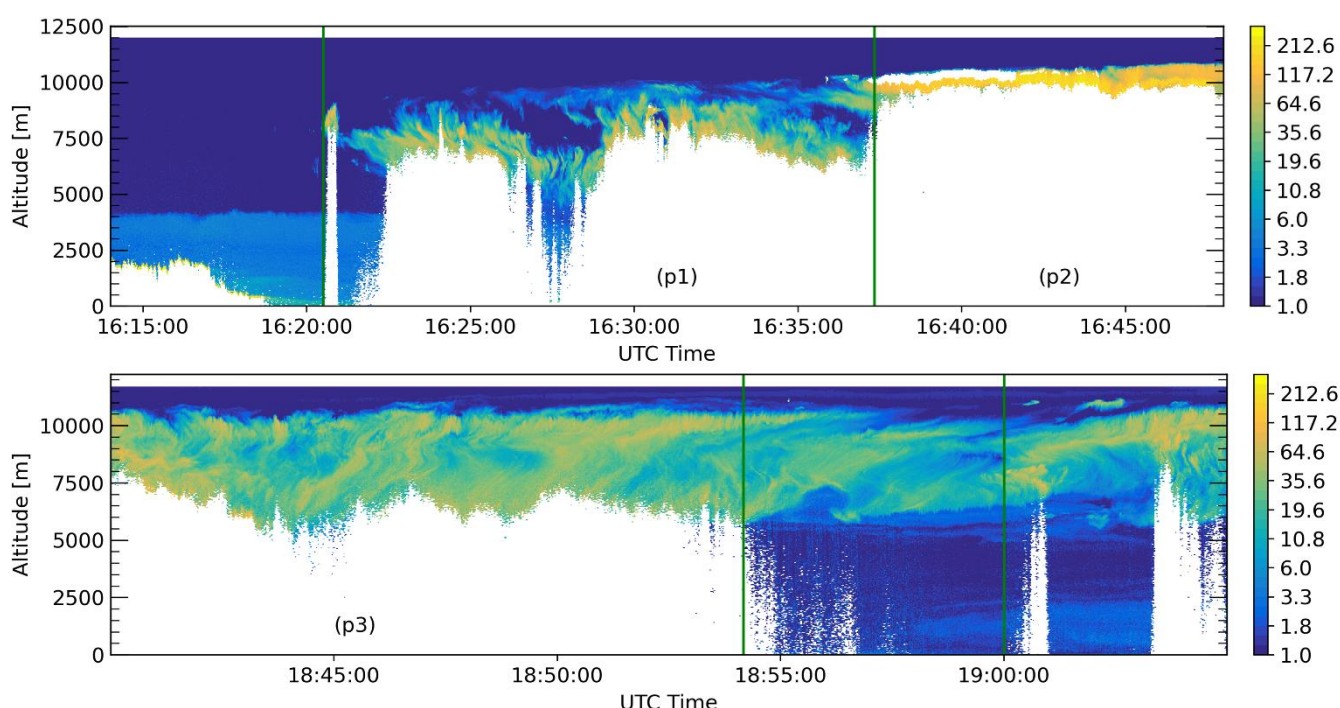


**Figure 6: Backscatter Ratio (BSR) measured by WALES Lidar at 532nm (color-coded). Two lidar legs performed during Mission 6 of the ML-Cirrus campaign on 29/03/2014 over the West Mediterranean, depicting a mesoscale convective system. P1 contains two towering convective cells. P2 contains the main convective cell. P3: depicts the outflow of the system. The white regions at cloud top are a result of saturation of the detector. The clouds in p1 and especially p2 are not fully depicted and only the top**
**portions of the cloud could be measured by the lidar system.**

The first special case consists of two lidar legs conducted during Mission 6 of the campaign on 29/03/2014 over the West Mediterranean (Voigt et al., 2017, Their Table 3). In these legs a multicell thunderstorm evolving into a Mesoscale Convective System (MCS) was measured. The flight was conducted in such a way that young, mature and dissipating clouds





were measured. In Figure 6, we present the BSR measured during these legs. We further split the legs into three parts, each

containing clouds on a different evolutionary stage, young cells (p1), mature cell (p2) and dissipating cell (p3). Between the

two lidar legs, in-situ measurements were performed, causing the time gap between parts 2 and 3.

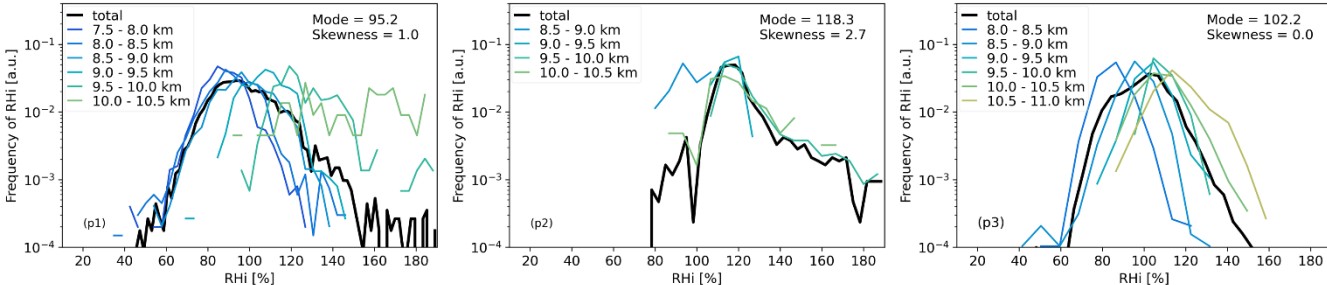

**Figure 7: PDF of in-cloud RHi in vertical layers with 500m depths from cloud bottom to cloud top. From left to right p1-Young cells, p2-Mature cell and p3-Stable cell, in accordance to Figure 6. The number of layers differs depending on the depth of the**

**cloud.**

Figure 7 contains the frequency distributions of RHi for the three parts. In order to study the altitude dependence of RHi

within these clouds we divide them into vertical layers with a depth of 500 meters. For the first part of the thunderstorm (p1),

the young cells, the mode of the RHi is 95 % and the skewness is 1. Regarding the vertical structure, the topmost layers of

these clouds are dominated by very high supersaturations. Moving deeper into the cloud the RHi values gradually drop to

saturation. For the second part (p2) the mode of RHi is 118 % and the skewness is 2.7. The majority of data points is

supersaturated. RHi reaches values higher than 180 % and is only rarely below saturation. Such high RHi values that exceed

the HOM threshold could be explained as a result of the uncertainties of the temperature data from ECMWF. Vertically only

a few levels could be measured. They all share similar characteristics. The third part of this case (p3), the stable cell has a

RHi mode of 102 % and no skewness. Subsaturated and supersaturated values of RHi are almost balanced through the cloud.

Vertically all layers have a similar shape of distribution. The mode of each layer decreases from cloud top to cloud bottom.

**3.3.2 Warm Conveyor Belt**

The second special case is a lidar leg, conducted over the United Kingdom and Belgium, as part of Mission 14 of the

campaign on 11/04/2014 (Voigt et al., 2017, their Table 3). The targeted system was the outflow of a warm conveyor belt

(WCB). For this leg a quasi-Lagrangian approach was selected, were the flight path of the aircraft coincided with the

movement and evolution of the cloud system, by following the geopotential lines of the WCB.





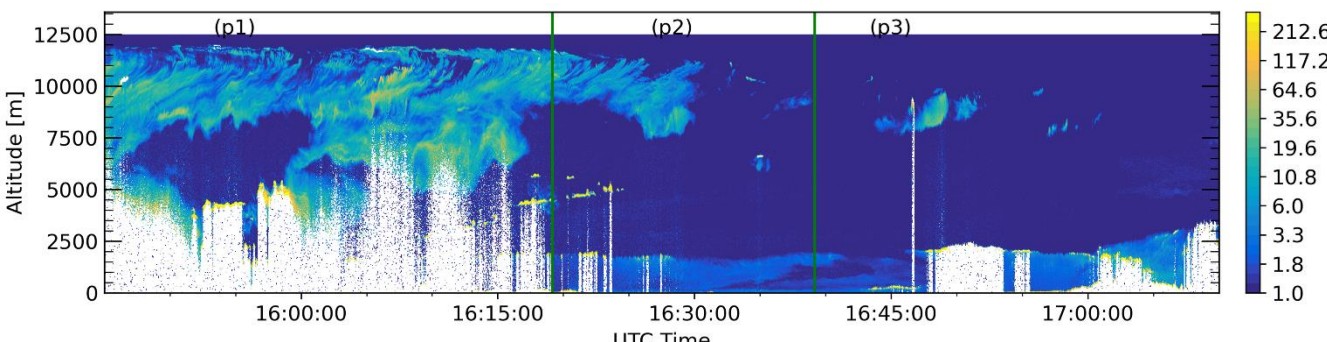

**Figure 8: Backscatter Ratio (BSR) measured by WALES Lidar at 532nm (color-coded). Measurement performed during Mission 14 of the ML-Cirrus campaign on 11/04/2014 over the United Kingdom and Belgium, depicting the outflow of a Warm Conveyor Belt (WCB). The flight was conducted in a semi-Lagrangian manner, following the axis of the evolution of the system. P1, P2 and P3 depict clouds in different phases of aging from young to old, respectively.**

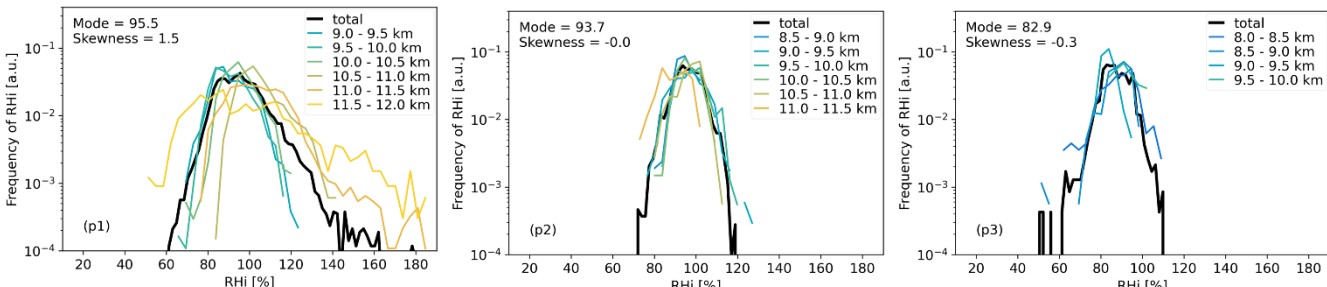

**Figure 9: PDF of in-cloud RHi in vertical layers with 500m depths from cloud bottom to cloud top. From left to right p1-Young phase, p2-Mature phase and p3-Stable phase, in accordance to Figure 8. The number of layers differs depending on the depth of the cloud.**

In Figure 8, we plot the BSR of the WCB. We once again split the case into three parts, depending on the evolutionary stage of the clouds and divide the clouds into vertical layers with a depth of 500m. For this lidar leg, HALO flew along the WCB from near the center of the system to its outer edge. Thus, we consider the first part to be the youngest. It has an RHi mode of 95.5 % and a positive skewness of 1.5. The range of the distributions reaches RHi values above 140 %. The vertical structure is rather uniform. The uppermost layers of the cloud seem to have more datapoints with higher supersaturations. The second part is slightly older. The mode here is 94 and the skewness close to 0. At this stage the cloud is stable. This is also seen in the vertical structure which is uniform throughout the cloud. The third part is the furthest from the center of the system and thus, we consider it to be the oldest of the three. The RHi mode for this cloud is 83 % and the skewness is slightly negative and the cloud has started dissipating at this stage (Figure 9).

## 4 Conclusions & Discussion

Using lidar measurements of cirrus clouds over the mid-latitudes and following the abovementioned methods, we found that the distribution of RHi in mid-latitude cirrus clouds has a 96 % mode value, which is close to the ice saturation threshold,





considering the error of 10-15 % in the calculation of RHi. Around 34 % of the in-cloud data points showed ice supersaturation; 3.8 % had RHi values higher than 120 % and 0.5 % exceed 140 % RHi. Ovarlez et al. (2002), Ström et al. (2003), Comstock et al. (2004), Gensch et al. (2008), Kübbeler et al. (2011), Petzold et al. (2017), Kaufmann et al. (2018)

and Krämer et al. (2009, 2020) also found an RHi mode close to or at ice saturation for mid-latitude cirrus clouds. Groß et al. (2014) analysed water vapor measurements by WALES taken over Germany, during November 2010 and found an RHi mode of 98 %, and an ice supersaturation frequency of 30 %, which is in accordance with our findings, but contrary to our findings they reported only 2 % of data points over RHi 120 %. Differences may arise due to the seasonal variability of RHi, and the fact that they performed measurements only in one atmospheric system (Kahn et al., 2008; Dzambo and Turner,

2016). Other studies also found supersaturations around 30 % in mid-latitude cirrus clouds. Namely, Comstock et al. (2004) analysed Raman lidar measurements taken over the Southern Great Plains (SGP) and Ovarlez et al. (2002) in-situ measurements over Scotland, during the INCA (INter hemispheric difference in Cirrus properties from Anthropogenic emissions) campaign (Ström et al., 2003). Only Jensen et al. (2001) found a higher ice supersaturation frequency, 49 %, but from in-situ measurements taken during the SUCCESS aircraft field campaign over an area close to SGP. It must be noted

that the SUCCESS campaign focused in measuring areas with high ice supersaturations.

After studying the correlation of RHi with temperature we found that, the probability for occurrence of supersaturation is slightly higher in the temperature range between 215 K and 225 K. Comstock et al. (2004), Krämer et al. (2009, 2020) and Jensen et al. (2013) also detected a trend to higher supersaturations for colder temperatures. Most of the supersaturated data points are in the HET regime; some are also found in the HOM regime and very few are even above water saturation, in

contrast to Krämer et al. (2009) who found no supersaturations above that threshold. The errors on the RHi values that stem from the used datasets should also be regarded.

Regarding the vertical structure of RHi, we found that the cloud tops mostly consist of data points with high supersaturations and RHi values above 140 %. Lower supersaturations gradually become dominant in deeper layers of the clouds, while the cloud bases are mostly subsaturated. This leads to the conclusion that ice crystals form near the cloud tops,

grow at cloud middle and sublimate or sediment at the cloud base. Model simulations by Heymsfield and Miloshevich (1995) and Spichtinger and Gierens (2009), predicted the same vertical distribution of supersaturation. Other field studies over the mid-latitudes also confirm our findings. Comstock et al. (2004) and Di Girolamo et al. (2009) used ground-based Raman lidar over the SGP and over Italy (EAQUATE Experiment), respectively, and noted higher ice supersaturation near cloud-top and sub saturation near cloud base. Sakai et al. (2014) measured cirrus clouds with an instrumented balloon and

ground-based lidar and also detected their highest supersaturations near the cloud tops. Dzambo and Turner (2016), also came to the same conclusions by using a combination of radiosonde and millimeter-wavelength cloud radar data from the SGP, as did Korolev and Isaac (2006) by using in-situ measurements.

In some cases, the uppermost cloud layer had lower supersaturations than deeper layers or was even subsaturated. The analysis of those cases showed that these clouds also had a lower RHi mode and a low or negative RHi skewness, leading to



the conclusion that they were older clouds in which ice crystal formation had ceased. Groß et al. (2014) also noted this behaviour in one of their analysed cases and came to the same conclusion.

      The cloud-free air in the vicinity of the measured cirrus clouds has a RHi mode of 68 %. Voigt et al. (2010) also find the cloud-free RHi values to be mostly subsaturated, but higher than 70 % RHi. Supersaturation with respect to ice is detected for 6.8 % of the cloud-free data points, although only 1.2 % have RHi over 120 %. Colder data points have a higher

probability of being supersaturated. Comstock et al. (2004) detected a higher supersaturation frequency for the cloud free air (9 %). Aircraft-based studies such as Heymsfield et al. (1998), Gierens et al. (1999, 2000) and Jensen et al. (2001, 2005), also detected high supersaturations over ice in the upper troposphere of the mid-latitudes. Most of the supersaturated data points are in the HET regime, although very few are also detected close or in the HOM regime. This was also detected by Ovarlez et al. (2002), Krämer et al. (2009, 2020) and Cziczo et al. (2013). The high probability density for warm data points

with RHi close to zero stems from measurements taken in the stratosphere, as we found by a closer analysis of individual cases.

      The correlation of RHi with temperature showed that mid-latitude cirrus clouds are most frequently detected in two domains. One around 225 K and close to saturation, RHi = 100 %, and a second one around 215 K which is subsaturated, RHi = 90 %. These two domains coincide with the two cirrus groups, in-situ and liquid-origin. In-situ represent the colder,

subsaturated domain and liquid-origin the warmer, saturated domain, as was also shown by Gasparini et al. (2018). More specifically, 30.8 % of the data points of in-situ formed clouds were supersaturated; mostly below the low HET threshold. On the other hand, liquid-origin clouds have a supersaturation occurrence of 36 % and supersaturation reaches the high HET threshold. The differences between the two groups are most pronounced for RHi up to 120 %; for higher values they have similar occurrence frequencies. Cloud-free air in the vicinity of liquid-origin cirrus clouds is also more likely to be

supersaturated.

      A variety of factors are able to affect the cloud formation and evolution. The ambient updraft is one of the most important. The intensity of the updraft is one of the regulating factors, that determines the cooling rate, supersaturation and nucleation process. In order to investigate the possible effect of the updraft on the two cloud types, we calculated the frequency distributions of the vertical velocities. We used vertical velocities from the ERA5 reanalysis dataset and

collocated them to the measured clouds. For both groups the mode value is very close to 0. The skewness of the distribution for the in-situ clouds is 0.4 and for the liquid-origin clouds it is -0.31 (not shown). Liquid-origin clouds are thus more frequently measured in environments with ever so slightly lower updrafts, than in-situ clouds.

      From the analysis of the two special cases, we identify three evolutionary stages, with distinctive characteristics. Young, newly formed clouds have an RHi mode close to ice saturation and an RHi PDF with positive skewness, resembling

a Rayleigh distribution. Skewness is higher for the uppermost layers, where more supersaturated data points are located and new ice crystals form. Mature clouds have a supersaturated RHi mode and an RHi PDF with skewness towards higher supersaturations, also close to a Rayleigh distribution. The vertical RHi distribution is rather uniform throughout the cloud, with slightly higher supersaturations near cloud top. Finally, stable clouds have an RHi mode close to saturation or





subsaturated and an RHi PDF with skewness close to zero, described by a nearly Gaussian distribution. Their vertical structure is rather uniform, although supersaturation might still be frequent near the cloud top. At this stage the clouds may still produce new ice crystals, but are nearing their dissipation, especially if the RHi mode is subsaturated. It should be noted, that especially for younger clouds most commonly only the upper layers could be measured with the lidar. Errors in the calculation of RHi also stem from the error of the ECMWF temperatures as discussed by Groß et al. (2014) and Kaufmann et al. (2018). Comstock et al. (2004) also chose a case and studied the behaviour of RHi during the evolution of a cirrus cloud.

They note similar stages in their life cycle as we do. Ovarlez et al. (2002), split their observed clouds into warm and cold regime with the threshold temperature being -40 ∘C. They found that a Gaussian distribution can be fitted over the warm clouds and a Rayleigh over the cold ones, due to a positive skewness towards higher RHi values.  They conclude, that this difference stems from the variability in the amount of time that is needed for a cirrus cloud to reach a stable state, where depositional growth and sublimation of ice crystals are in equilibrium (Khvorostyanov and Sassen, 1998; Hoyle et al., 2005).

They interpreted the positive skewness as being an indication of clouds that have not yet reached maturity. This is also observed in our study. Spichtinger et al. (2004) detected positive skewness in the RHi distributions in the higher/colder parts of the clouds measured during MOZAIC (Measurement of Ozone by Airbus In-service aircraft project). They also concluded that this observation is relevant to the relaxation times and the effect vertical motion has on them (Gierens, 2003). Finally, Groß et al. (2014), used the shape of the RHi distribution in order to draw conclusions about the evolutionary stage of the

cloud in their case study.

## 5 Summary and Outlook

In this study we combined water vapor measurements by the WALES lidar system, conducted during the ML-Cirrus campaign and model temperatures by ECMWF and studied the general characteristics, vertical structure and temporal evolution of humidity related to saturation within mid-latitude cirrus clouds and their adjacent cloud-free air. The German

research aircraft HALO, employed during this campaign, is capable of reaching the altitudes needed for the study of cirrus clouds via lidar and has a long enough range allowing for continuous measurements of extended systems over land and ocean. The use of a lidar instrument is advantageous over in-situ measurements, as it provides a 2D curtain including the complete vertical structure of the measured clouds. This way one overflight is sufficient enough to measure a whole cloud or cloud system and get information about the whole atmospheric column from flight altitude until the ground for cloud-free

regions. The WALES lidar system is a unique and powerful instrument as it combines HSRL and DIAL measurement techniques. Thus, the optical properties of the cirrus clouds as well as water vapor concentrations are measured simultaneously with a high spatial and temporal resolution.  The use of collocated model temperatures from ECMWF does not negatively affect our analysis, despite the known lower resolution and errors. Thus, we are confident that our method is suitable and effective for the study of cirrus clouds in the future.





From our analysis, we detect high supersaturations —reaching the threshold for homogeneous nucleation— within the clouds, but also in the cloud free air around them. Within the cirrus clouds we detect a clear vertical structure of RHi. The uppermost parts of the clouds are mostly supersaturated with RHi frequently above 140 %. That is where new ice crystals form. In the cloud middle lower supersaturated RHi values are detected and the cloud base is most frequently subsaturated. We detect two regimes which we identify as the two cloud groups suggested by Krämer et al. (2016); in-situ formed and

liquid-origin cirrus. These two cloud groups have been shown to have different effects on the climate (Krämer et al., 2020). Krämer et al. (2016) and Gasparini et al. (2018) studied them by means of model simulations and Luebke et al. (2016) and Krämer et al. (2020) by in-situ and satellite measurements. To our knowledge this is the first study of these cloud groups with an airborne lidar, where we discuss the characteristics of RHi within them and at their adjacent environment. We also conclude, that the shape of the frequency distribution of RHi within cirrus clouds and the vertical structure of RHi can be

used as a signature indicating different evolutionary stages. Rayleigh distributions are indicative of young clouds or cloud layers where ice crystals are being formed —especially if the RHi mode is supersaturated— and Gaussian distributions of older clouds or cloud layers that have reached a mature state of equilibrium, or even dissipating clouds, if additionally, the RHi mode is subsaturated.

    The distributions of ice nucleating particles, water vapor and the ambient temperature have a dependence on latitude.

Because of that cirrus clouds in the mid- and high latitudes have different characteristics and different effects on the climate, as their microphysical and radiative effects strongly depend on their formation mechanism (Gensch et al., 2008; DeMott et al., 2010; Hong and Liu, 2015; Gasparini et al., 2018; Krämer et al., 2020). Lidar measurements of cirrus clouds in high latitudes are scarce and necessary in order to compare their characteristics with mid-latitude cirrus clouds and investigate possible differences.

**Data Availability**

The lidar measurements from the ML-Cirrus campaign, that were used in this study can be found at the HALO data base: https://halo-db.pa.op.dlr.de/mission/2. Modell temperatures are obtained from ECMWF (European Centre for Medium-Range Weather Forecasts).

**Author Contributions**

GD and SG conceptualized the study. Data curation by MW. Formal analysis, investigation and visualization by GD assisted by SG and MW. Calculation of backwards trajectories and cloud classification by CR and MK. Writing of original draft by GD under supervision of SG. Review and edit of the manuscript by SG, MW, MK and CR.



**Competing Interests**

Some of the authors are members of the editorial board of ACP.

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
