# Peer review of "Characteristics of supersaturation in mid-latitude cirrus clouds and their adjacent cloud-free air"

_Atmospheric Chemistry and Physics, 2022_

## Referee Comment (RC1)

Review of ACP MS No.: acp-2022-717
Title: Characteristics of supersaturation in mid-latitude cirrus clouds and their adjacent cloud-free air
Author(s): Georgios Dekoutsidis, Silke Groß, Martin Wirth, Martina Krämer, and Christian Rolf
MS type: Research article
Iteration: Correction

General Comments:

A longstanding question in cloud physics is whether homogeneous ice nucleation (henceforth hom) is a relevant process in the atmosphere.  Recent satellite remote sensing studies (Sourdeval et al., 2018, ACP; Gryspeerdt et al., 2018, ACP; Mitchell et al., 2018, ACP) have provided evidence that hom can strongly affect the microphysics of cirrus clouds as inferred through changes in the relative concentration of ice particles, but direct in situ airborne measurements have not yet provided compelling evidence for this.  This study by Dekoutsidis et al. appears to be the first measurement-based study that provides mechanistic evidence that hom is an important process affecting cirrus cloud properties.  That is, by profiling the atmosphere during the ML-Cirrus campaign with the WALES lidar system to obtain the relative humidity over ice (RHi) inside and outside of cirrus clouds, this study reveals the cloud levels relative to cloud top where hom is likely to dominate, and cloud levels where heterogeneous ice nucleation (i.e., het) or sublimation is likely to be the dominant process.  While previous work has shown that RHi in cirrus is highest near cloud top (Diao et al., 2016, JGR), no earlier study has shown that RHi conducive to hom is typically found near cirrus cloud top.  This study thus provides a mechanistic foundation helpful for interpretating other cirrus cloud studies, including anvil cirrus as shown in Fig. 7 of this manuscript.

Figure 10 in Mitchell et al. (2018, ACP) shows CALIPSO retrieved $N_i$ for cirrus cloud layers at various temperature levels in terms of their layer thickness, where layer thickness is characterized by the difference $T_c - T_{top}$ with $T_c$ = median cloud radiative temperature and $T_{top}$ = cloud top temperature.  The cirrus clouds sampled were relatively thick optically, having optical depths (OD) ranging from ~ 0.3 to 3.0.  A key finding was that the relatively thin cirrus cloud layers had a much higher $N_i$ than the other cirrus cloud layers.  This begs the question "What is causing the high $N_i$ in the thin cirrus layers?"  Would it be possible for the authors to organize their RHi data in terms of cirrus cloud layer geometrical thickness (for a similar OD range) in order to address this science question?  This is similar to what has already been done, but in this case RHi is not being profiled within a given cloud, but rather a single median RHi characterizes each cloud layer.  The working hypothesis to be tested might be that relatively thin cirrus have RHi closer to the hom RHi threshold than for thicker cirrus clouds.  Theory supporting this hypothesis may be found in Spichtinger and Geirens (2009, ACP).

I found little to criticize in this manuscript, which is well written and organized with high quality figures. A few comments are given below for possible improvements. This manuscript is definitely worthy of publication in ACP.

Major Comments:

1. Lines 204-205: Do you think this may be due to entrainment?

2. Lines 242-250: The probability density at cloud top for RHi > 140% is ~ 0.012 in Fig. 3 (for all clouds sampled) but is much lower at cloud top for both in situ and liquid origin cirrus clouds in Fig. 5. Perhaps this is an artefact of different binning procedures, but some explanation seems warranted.

3. Lines 329-330: This conclusion also appears consistent with the findings in Diao et al. (2015, JGR) that use RHi and ice crystal concentration measurements to define 5 stages of cirrus cloud evolution (Diao et al., 2013, GRL), with the ice nucleation stage in the uppermost portion of cirrus clouds. Please cite these papers if appropriate.

4. Lines 352-354: From Fig. 4, these two maxima (corresponding to liquid origin and in situ cirrus) appear to occur at 223 K and 217 K, respectively. Please check this and revise the maxima temperatures if this is correct.

5. Lines 394-395: What was the flight ceiling for HALO during ML Cirrus? Please report this in the paper.

6. Lines 419-422: The global cirrus cloud retrievals of $N_i$ and $D_e$ reported in Mitchell et al. (2018, ACP; Figs. 9 & 11) also show considerable differences between tropical, mid- and high latitude cirrus cloud properties as a function of season. Sourdeval et al. (2018, ACP) shows the same for $N_i$. Please cite these studies if appropriate.

Technical Comments:

1. Line 109: where => were?

2. Line 127: No comma is needed in this sentence.

---

## Referee Comment (RC2)

**Review on Dekoutsidis et al. paper**

This is an excellent study, with a substantial contribution to scientific progress within the scope of ACP. It is based on airborne lidar observations from the PM-Cirrus experimental campaign in the broader Europe domain, and provides characteristics of supersaturation conditions in and around cirrus clouds. The authors classify their finding to clouds formed from in-situ or liquid-origin processes, and provide findings on the relative humidity conditions observed in different locations of the cloud and at different phases of a cloud evolution. The paper is well written and the overall presentation is well structured and clear. It is recommended for publication in ACP after a few revisions provided bellow.

**General comment:**

A general question popup in my mind while reading the manuscript, which would be nice to have as a reader: What are the cloud depths observed for the cloud categories "in-situ" and "liquid-cloud" in the dataset discussed here? If the authors can provide this information (e.g. median and standard deviation of the cloud depths in the 2 categories) it would be nice.

**Specific comment:**

**Page 3, line 95:** "…accurate and suitable for the study of ice clouds: Groß et al. (2014), used water vapor measurements from WALES and found a good agreement with in-situ measurements. Kiemle et al. (2008) …".

This sentence raise the question if these authors provide an indicative quantification of the good agreement from this study, and if yes what was the result. I suggest for the authors to revise this part in order to highlight the new information provided in Groß et al. (2014) on the accuracy of the measurements in ice clouds, in relation to the Kiemle et al. (2008) study discussed afterwords.

**Page 4, line 117:** "Groß et al. (2014) compared the model temperature with in-situ measurements and found that ECMWF temperature data induces an error of about 10 to 15 % in the calculated RHi. Despite that, they concluded that the ECMWF model temperature is suitable for the study of cirrus clouds at the mid-latitudes".

It would be nice to include a comment related to which extend this error is representative for the cirrus clouds of this study. Furthermore, do you find similar induced errors in the calculated RHi during the ML-Cirrus campaign (e.g. comparing the temperatures from the radiosondes and the ECMWF data)? Is the expected uncertainty constant for the different temperatures (i.e. for 230 K and 215 K), or higher uncertainties are expected for lower/higher temperatures?

**Page 4, lines 105 – 108:** It I not clear how the classification between the in-situ formed and the liquid origin clouds is done for this study. It would be nice if the authors can provide more specific information on how this classification was done for this work.

**Page 5, line 130:** "We consider the vicinity around cirrus clouds as a maximum horizontal distance of 250 km from the cloud edges and altitudes from 7 km to 12 km as we mostly detect cirrus clouds in this range".

Will the results of RHi in cloud-free areas change as the distance from the cloud shortens, i.e. in the cloud twilight zone? It would be interesting to use these data and investigate if/how the RHi changes in the clouds twilight zone. it would be nice if the authors can include a comment on that.

**Page 5, lines 134 - 141:** "Finally… abundant".

Please rephrase to make this part more clear on which are the thresholds used and what they represent. Specifically, in the first sentence is not clear on which parameters are the thresholds calculated. For the 3rd sentence on, is not clear which is the physical rationale that the specific temperature thresholds represent/imply.

**Page 5, line 147:** "In order to get a more detailed insight in the supersaturation we define three bins of RHi, 100 %–120 %, 120 %–140 % and > 140 %. RHi 120 % and 140 % can be considered approximate thresholds for HET and HOM respectively".

Similarly as for the previous comment, it would be good to rephrase this sentence to provide a clearer connection between the HET and HOM relevant temperature regions mentioned and the three defined bins.

**Page 5, line 170 – Page 7, line 186:** I think it would be nicer for the reader if you could combine the discussion of these 3 paragraphs, by mentioning the general statistics along with the qualitative discussion in the 2 last paragraphs.

**Page 6, line 170:** "The in-cloud data points reach temperatures down to 207 K and are most frequently detected close to ice saturation (RHi = 100 %) for the entire temperature range."

From figure 2 it seems like the majority of the in-cloud points at T > 230 K are between RHi = 75% to 100%, and well below 100%. Can you comment on this?

**Page 6, line 170:** "This feature is detected in many cases and is indicative of air masses with various temperatures and a constant water vapor mixing ratio around 1.5ppmv, which is the minimum value observed in the upper troposphere (Krämer et al., 2009) or data points measured in the stratosphere."

Please rephrase this part to make clearer for which ranges in troposphere and stratosphere this mixing ration is observed.

**Page 7, line 202:** "In summary, high supersaturations are detected at cloud top and lower supersaturations gradually become dominant towards the middle of the clouds. From around the mid-point of the clouds and until the cloud bottom most data points are subsaturated."

In which extend could the attenuation of the lidar signal through the cloud affect the RHi values closer to the cloud bottom? Could this contribute to the statistics presented in these altitudes? If possible it would be interested to comment on this in the manuscript.

**Page 9, Line 204:** "For around 20 % of the cases the uppermost layer has lower supersaturation than deeper layers or is even subsaturated (not shown)."
Can you comment if these clouds are observed with some characteristic that differentiate them from the majority? E.g. temperature or vertical extend or aerosol abundance..?

**Section 3.2:** Similar comment as earlier. I think it would be nicer for the reader if you could combine the discussion of these paragraphs, by mentioning the relevant general statistics of Table 2 along with the qualitative discussion of Figure 4.

**Page 10, Line 252:** "we choose two special cases".
If possible indicate how the 2 cases chosen are special in compared to the rest of the dataset.

**Figure 6, 8:** Please include legents in the plot's colorscales. And values (or mention in the legent if unitless).

**Figure 7:** "The number of layers differs depending on the depth of the cloud".
This is also depending on the lidar penetration in the cloud (especially for p2 phase). Consider revising this sentence accordingly.

**Page 13, Line 308:** "… Gensch et al. (2008), Kübbeler et al. (2011), Petzold et al. (2017), Kaufmann et al. (2018), and Krämer et al. (2009, 2020) also found an RHi mode close to or at ice saturation for mid-latitude cirrus clouds".
It would be nice to mention the type of dataset used for these studies, i.e. in-situ measurements, radiosondes, lidars?

**Page 14, Line 342:** "Voigt et al. (2010) also find the cloud-free RHi values to be mostly subsaturated, but higher than 70 % RHi".
Please improve the syntaxes of this sentence to make it clearer.

**Page 14, Line 366:** "Liquid-origin clouds are thus more frequently measured in environments with ever so slightly lower updrafts, than in-situ clouds".
One would expect that from the way the liquid-origin clouds were defined in section 2.3 they would be in environments with stronger vertical motions (updrafts and downdrafts in the cloud) than the in-situ clouds, while the sentence is arguing the opposite. If possible, it would be nice if the authors can enhance this discussion (e.g. by adding some other references supporting this statement) or provide a comment on why this may be observed.

**Page 15, line 397:** "The use of a lidar instrument is advantageous over in-situ measurements, as it provides a 2D curtain including the complete vertical structure of the measured clouds. This way one overflight is sufficient enough to measure a whole cloud or cloud system..".
The lidar signal gets often totally attenuated as it measures through a cloud (which is also evident in the 2 case studies of this work). It would be better if the authors rephrase this part to be more accurate to the real capabilities of a lidar instrument.

**Page 16, Line 422:** "Lidar measurements of cirrus clouds in high latitudes are scarce and necessary in order to compare their characteristics with mid-latitude cirrus clouds and investigate possible differences".
Lidar measurements of cirrus clouds in high latitudes are not scarce, considering the 15-year record of CALIPSO dataset. But the lidar water vapor measurements are. Consider rephrasing this part.

**Technical corrections:**
Page 1, line 16: "..ML-Cirrus data-set**,** there are two.."
Page 10, line 250: "..lower than **the ones in** the deeper.."
Page 11, line 273: "..**in** the young cells..."
Page 11, line 278: "**In** the third part.."
Page 14, line 369: "Young, newly formed**,** clouds.."

---

## Author Comment (AC1)

**Referee #1**

Dear Referee 1,

we would like to thank you for your time and the valuable feedback. Your positive opening comments are a strong motivation. Your suggestion to group the RHi data depending on the geometrical depth of the layers has stirred discussions in the group and we find it a very interesting scientific question.

Regarding the reviews, please find in the following a detailed reply on your comments. We present your comments/questions in Bolt lettering followed by the reply. For changes in the text I present the original text in blue and the updated/added text in green color.

**Major Comments**

1. **Lines 204-205: Do you think this may be due to entrainment?**

   Yes, entrainment could be a possible explanation for this finding. Unfortunately, entrainment is not something we can study with a lidar instrument. Another possibility is that this stems from measurements of clouds in their last stages of their life cycle. Since we perform a statistical analysis of all the data it is not possible to attribute this to certain clouds. Finally, another probable explanation would be that the cloud mask misidentifies some areas as in-cloud in regions where the cloud-top is not smooth, leading to some otherwise lower-RHi cloud-free points to be counted with the in-cloud points

2. **Lines 242-250: The probability density at cloud top for RHi > 140% is ~ 0.012 in Fig. 3(for all clouds sampled) but is much lower at cloud top for both in situ and liquid origin cirrus clouds in Fig. 5. Perhaps this is an artefact of different binning procedures, but some explanation seems warranted.**

   Thank you for pointing this out. This confusing characteristic is not due to binning, but rather a different handling of the data set when treated as a whole and in groups. When analyzing the whole dataset, in-situ and liquid-origin separation is not considered. In some cases, the uppermost part of a detected cloud is classified as in-situ while beneath is a liquid-origin part. For the analysis of
 all data points this was seen as one cloud. We have repeated the analysis for all clouds, separating already at that step the liquid-origin and in-situ part. We have recreated the plot and changed the text accordingly.

3. **Lines 329-330: This conclusion also appears consistent with the findings in Diao et al. (2015, JGR) that use RHi and ice crystal concentration measurements to define 5 stages of cirrus cloud evolution (Diao et al., 2013, GRL), with the ice nucleation stage in the uppermost portion of cirrus clouds. Please cite these papers if appropriate.**

Thank you for bringing these publications to our attention. Diao et al., 2015 and 2013 also classify the ice nucleation, growth and sublimation stages based mainly on the values of Relative Humidity over ice. We cite the papers as follows:

…sediment at the cloud base. Diao et al., 2013 & 2015 also define the ice nucleation, growth and sublimation stages based on the measured ice super- and subsaturation. Model simulations…

Regarding the vertical structure, Diao et al., 2015 consider ice supersaturation with respect to distance from tropopause rather than cloud top. We believe this would be misleading for the reader as we define our vertical structure from cloud top. Despite that, Diao et al., 2013 classify evolutionary phases which largely agree with our conclusions on cirrus evolution. Based on this we cite these papers also at line 380 as follows:

…cycle as we do. The evolutionary stages we find are also confirmed by in-situ measurements (Diao et al., 2013 & 2015). Ovarlez et al. (2002), split their observed…

4. **Lines 352-354: From Fig. 4, these two maxima (corresponding to liquid origin and insitu cirrus) appear to occur at 223 K and 217 K, respectively. Please check this and revise the maxima temperatures if this is correct.**

Thank you for noticing this in the plots. We have checked the histograms with the latest binning used in the paper and we find the maximum for liquid origin at 225 K 100% RHi and for in-situ at 218 K 79% RHi. The text has been updated accordingly at lines 17, 172, 228, 353. Line 172:

…distribution is bimodal: One peak can be seen at a temperature of 225K and ice saturation, RHi 100%, and a second one at 218 K and below ice saturation at RHi around 79 %. An increase…

5. **Lines 394-395: What was the flight ceiling for HALO during ML Cirrus? Please report this in the paper.**

Unfortunately, there is no definite answer to this request as there was no fixed flight ceiling set. According to Krautstrunk and Giez 2012 the maximum cruise altitude for HALO is 15540m. After HALO was configured for the ML-CIRRUS campaign and the extra weight was added from the instrumentation and fuel this height was not achieved during the campaign. The flight planning was done in such a way that HALO would fly at least 1.5km over the cloud tops in order to warrant good measurements with the lidar. We have added the extra information at section 2.1 as follows:
Line 86:

…research flights over Central Europe and the NE Atlantic. One advantage of HALO is its high flight ceiling of 15 km (Krautstrunk and Giez, 2012). Due to its payload during the research flights this altitude was not reached, but the flight planning was done in such a way that HALO

would fly at least 1.5km over the cloud tops in order to warrant good measurements with the lidar. More details…

6. **Lines 419-422: The global cirrus cloud retrievals of $N_i$ and $D_e$ reported in Mitchell et al.(2018, ACP; Figs. 9 & 11) also show considerable differences between tropical, mid and high latitude cirrus cloud properties as a function of season. Sourdeval et al.(2018, ACP) shows the same for $N_i$. Please cite these studies if appropriate.**

The two studies are relevant to be cited and we thank you for your suggestion. The paragraph starting at line 419 has been rewritten as follows:

Ice nucleating particles, water vapor and the ambient temperature are three factors that strongly affect the formation of cirrus clouds. Common among these three, is that they have a dependence on latitude (DeMott et al., 2010). The microphysical characteristics and thus the radiative effects of cirrus clouds strongly depend on their formation mechanism (Gensch et al., 2008). Based on this it is known that cirrus clouds in the mid- and high latitudes also have different characteristics (Hong and Liu, 2015; Gasparini et al., 2018; Mitchell et al., 2018; Sourdeval et al., 2018) and different effects on the climate (Hong and Liu, 2015; Mitchell et al., 2018; Krämer et al., 2020). Lidar measurements of cirrus clouds in high latitudes are scarce and necessary in order to compare their characteristics with mid-latitude cirrus clouds and investigate possible differences.

**Technical Comments**

1. Line 109: where => were?
2. Line 127: No comma is needed in this sentence.

Both technical comments have been corrected in the text.

---

## Author Comment (AC2)

**Referee #2**

Dear Dr. Marinou,

we would like to thank you for investing your time to review our submitted paper and also for your constructive feedback.

Please find in the following a detailed reply on your comments. I present your comments/questions in Bolt lettering followed by the reply. For changes in the text I present the original text in blue and the updated/added text in green color.

**General Comment**

1. **What are the cloud depths observed for the cloud categories "in-situ" and "liquid-cloud" in the dataset discussed here? If the authors can provide this information (e.g. median and standard deviation of the cloud depths in the 2 categories) it would be nice.**

   Thank you for pointing this out, as it is something we had not considered. I have done the calculations for the mean depths of the cloud categories. Following text has been added to section 2.3 Cirrus Classification:

   …was split and grouped accordingly. For a more complete understanding of the two cloud types, the cloud depths for each group have been calculated. The In-situ clouds in our dataset have a mean cloud depth of 983 m with a standard deviation of 500 m and the liquid-origin clouds a mean depth of 1255 m with a standard deviation of 592 m.

**Major Comments**

1. **Page 3, line 95:** "…accurate and suitable for the study of ice clouds: Groß et al. (2014), used water vapor measurements from WALES and found a good agreement with in-situ measurements. Kiemle et al. (2008) …".
   **This sentence raise the question if these authors provide an indicative quantification of the good agreement from this study, and if yes what was the result. I suggest for the authors to revise this part in order to highlight the new information provided in Groß et al. (2014) on the accuracy of the measurements in ice clouds, in relation to the Kiemle et al. (2008) study discussed afterwords.**

   We have revised this section as follows, in order to describe in more detail the comparison between water vapor measurements from WALES and in-situ instrumentation conducted by Groß et al., 2014:

   …DIAL technique. Regarding the errors of the measurements, Kiemle et al. (2008) estimated the statistical error of the water vapor retrieval to be about 5 %, although the exact value is dependent on various parameters that differ for individual measurements.

Errors that arise due to the high spatial inhomogeneity of the backscatter within cirrus clouds are kept below 5 % by filtering. Finally, the Rayleigh-Doppler effect is corrected in the retrieval algorithm, leaving an error of less than 2 % (Groß et al., 2014). Groß et al. (2014), compared water vapor DIAL measurements from WALES, taken during a flight of the HALO Techno-mission with simultaneous in-situ measurements taken from an aircraft flying below HALO (their Figure 4). They found the measurements to be in a good agreement with a deviation of <1 % for the time periods where the two aircraft where on close horizontal distances. They also note the capability of the WALES-DIAL measurements to resolve even small-scale features. This leads to the conclusion that, WALES-DIAL measurements are accurate and suitable for the study of ice clouds (Groß et al., 2014). In depth description…

2. **Page 4, line 117:** "Groß et al. (2014) compared the model temperature with in-situ measurements and found that ECMWF temperature data induces an error of about 10 to 15 % in the calculated RHi. Despite that, they concluded that the ECMWF model temperature is suitable for the study of cirrus clouds at the mid-latitudes".
**It would be nice to include a comment related to which extend this error is representative for the cirrus clouds of this study. Furthermore, do you find similar induced errors in the calculated RHi during the ML-Cirrus campaign (e.g. comparing the temperatures from the radiosondes and the ECMWF data)? Is the expected uncertainty constant for the different temperatures (i.e. for 230 K and 215 K), or higher uncertainties are expected for lower/higher temperatures?**

Groß et al. (2014) showed that the mean temperature difference between ECMWF temperature and temperature sensors on-board HALO was 0.8 K nearly independently from the height and estimated a resulting maximum relative uncertainty of 10 to 15 % for the calculated RHi at typical cirrus temperatures. They use water vapor measurements taken from WALES over the mid-latitudes and temperature fields from ECMWF. We use the same also in our study. The differences between their study and ours is the time of year as the Techno-mission was conducted in autumn and the ML-Cirrus Mission in spring. Based on this we believe that their error estimation is representative also for our study. We clarify this in the text as follows:

…and Urbanek et al. (2018). Groß et al. (2014) studied the applicability of the ECMWF temperature field for the calculation of RHi. They compared the ECMWF temperatures with those measured by in-situ sensors on-board HALO and found a difference of 0.8 K for the typical height and temperature range of cirrus clouds. This induces an error of about 10 to 15 % in the calculated RHi. Despite that, they concluded that the ECMWF model temperature is suitable for the study of cirrus clouds at the mid-latitudes. Since we use the same data and similar method we deem their findings to be representative also for our study.
In the next step…

3. **Page 4, lines 105 – 108: It I not clear how the classification between the in-situ formed and the liquid origin clouds is done for this study. It would be nice if the authors can provide more specific information on how this classification was done for this work.**

   We added the following text:

   …temperatures (< 235 K). In our study we use the classification from Luebke et al., 2016 and Krämer et al., 2016. They calculate 24-hour backwards trajectories. For the wind data they use the ECMWF reanalysis dataset ERA – Interim (ECMWF, 2011). For vertical transport, diabatic heating rates are used with the trajectory module of the Chemical Lagrangian Model of the Stratosphere (CLaMS) (McKenna et al., 2002). The ice water content (IWC) is then calculated with CLaMS-Ice. The clouds are then grouped depending on the simulated IWC along the track and the location of the maximum IWC value. For some…

4. **Page 5, line 130:** "We consider the vicinity around cirrus clouds as a maximum horizontal distance of 250 km from the cloud edges and altitudes from 7 km to 12 km as we mostly detect cirrus clouds in this range".
   **Will the results of RHi in cloud-free areas change as the distance from the cloud shortens, i.e. in the cloud twilight zone? It would be interesting to use these data and investigate if/how the RHi changes in the clouds twilight zone. it would be nice if the authors can include a comment on that.**

   Groß et al., 2014, Urbanek et al., 2017 and 2018, define the cloud edge at a BSR value of 4, 2 and 3 respectively. In our study we choose a threshold of 3 for the BSR. Both Groß et al., 2014 and Urbanek et al., 2018 point out that other values were also considered with no effect on the distribution of RHi.  The later claim that no difference was seen in a range from BSR = 2 to BSR = 25. The twilight zone of the clouds would consist of data points with a BSR slightly greater than the selected threshold and it was shown that there is no difference to the RHi there.

5. **Page 5, lines 134 - 141:** "Finally… abundant".
   **Please rephrase to make this part more clear on which are the thresholds used and what they represent. Specifically, in the first sentence is not clear on which parameters are the thresholds calculated. For the 3rd sentence on, is not clear which is the physical rationale that the specific temperature thresholds represent/imply.**

   The paragraph has been revised as follows:

   Finally, to get some insight on the microphysics of the clouds and the ice nucleation processes, we calculate three temperature dependent thresholds. Two for heterogeneous nucleation (HET), and one for homogeneous nucleation (HOM) (Urbanek et al., 2017, their Table 1, and original formulations from Krämer et al., 2016 ). We also calculate the ice and water saturation thresholds. The water saturation threshold is the limit above which water droplets can form in addition to ice crystals. In-situ HOM nucleation occurs when

supercooled solution droplets (SSP) are lifted up to altitudes with very low temperatures (<235 K). For HET nucleation to take place, ice nucleating particles (INP) are needed. Different INP have different freezing thresholds. We specify a high threshold where we consider inefficient INP which lead to higher RHi being necessary for the nucleation, and a low threshold for easily activated INP where lower RHi is necessary for nucleation. For the high threshold we choose coated soot as an example of an inefficient INP and for the low threshold we choose mineral dust which is more efficient as an INP and more abundant (Pruppacher and Klett, 1997; Kärcher and Lohmann, 2003; Gensch et al., 2008; Hoose and Möhler, 2012; Cziczo et al., 2013; Krämer et al., 2016; Ansmann et al., 2019).

6. **Page 5, line 147:** "In order to get a more detailed insight in the supersaturation we define three bins of RHi, 100 %–120 %, 120 %–140 % and > 140 %. RHi 120 % and 140 % can be considered approximate thresholds for HET and HOM respectively".
**Similarly as for the previous comment, it would be good to rephrase this sentence to provide a clearer connection between the HET and HOM relevant temperature regions mentioned and the three defined bins.**

We have revised this sentence as follows:

In the temperature range of our data, RHi 120% could be considered as an approximate threshold for HET nucleation and RHi 140% for HOM nucleation respectively (Koop et al., 2000; Haag et al., 2003; Comstock et al., 2004; Khvorostyanov and Curry, 2009; Kärcher, 2012). Based on that and in order to present our data in a more easily understandable way we define three bins of RHi, 100 %–120 %, 120 %–140 % and > 140 % representing the low HET, high HET, and HOM nucleation regimes respectively (See 2.4).

7. **Page 5, line 170 (151?) – Page 7, line 186: I think it would be nicer for the reader if you could combine the discussion of these 3 paragraphs, by mentioning the general statistics along with the qualitative discussion in the 2 last paragraphs.**

Thank you for your suggestion. We believed that the way we have structured this section makes it easily understandable for the reader and would like to keep it as is.

8. **Page 6, line 170:** "The in-cloud data points reach temperatures down to 207 K and are most frequently detected close to ice saturation (RHi = 100 %) for the entire temperature range." **From figure 2 it seems like the majority of the in-cloud points at T > 230 K are between RHi = 75% to 100%, and well below 100%. Can you comment on this?**

The mode value of RHi for the whole data set is 96%. For the in-situ and liquid-origin clouds the mode value happens to be the same, 96% RHi. For T > 230 K the peak is at 86 % RHi. This tail to lower RHi values for higher temperatures might be explained by the vertical structure of the clouds. In general, the warmer parts of the clouds are closer to the cloud bottom which we have shown is also more frequently subsaturated.

9. **Page 7, line 182:** "This feature is detected in many cases and is indicative of air masses with various temperatures and a constant water vapor mixing ratio around 1.5ppmv, which is the minimum value observed in the upper troposphere (Krämer et al., 2009) or data points measured in the stratosphere."
   **Please rephrase this part to make clearer for which ranges in troposphere and stratosphere this mixing ration is observed.**

   We have revised this phrase as follows:

   This feature is detected in many cases. For temperatures under 215 K it represents data points with a water vapor mixing ratio around 1.5ppmv, which is the minimum value observed in the upper troposphere (Krämer et al., 2009). For higher temperatures it most probably stems from data points measured in the stratosphere.

10. **Page 7, line 202:** "In summary, high supersaturations are detected at cloud top and lower supersaturations gradually become dominant towards the middle of the clouds. From around the mid-point of the clouds and until the cloud bottom most data points are subsaturated."
    **In which extend could the attenuation of the lidar signal through the cloud affect the RHi values closer to the cloud bottom? Could this contribute to the statistics presented in these altitudes? If possible it would be interested to comment on this in the manuscript.**

    The attenuation of the signal should not have an effect on the calculated RHi. A potential drawback would be the complete attenuation. In this case the cloud-bottom used in our calculations is not the actual base of the cloud, since we do not have measurements from that depth. We have addressed this as follows:

    …most data points are subsaturated. It should be noted that in some cases the lidar signal gets completely attenuated before reaching the actual cloud base. In our analysis we consider the altitude of complete attenuation as the cloud base. For around 20 %...

11. **Page 9, Line 204:** "For around 20 % of the cases the uppermost layer has lower supersaturation than deeper layers or is even subsaturated (not shown)."
    **Can you comment if these clouds are observed with some characteristic that differentiate them from the majority? E.g. temperature or vertical extend or aerosol abundance..?**

    During our analysis we studied these cases in detail. We found that this characteristic was present in in-situ as well as liquid-origin clouds measured under various atmospheric conditions. Unfortunately, no grouping of these clouds was possible. One possibility which

is also discussed by Groß et al., 2014 would be that these clouds are in a different evolutionary stage along their life cycle.

12. **Section 3.2: Similar comment as earlier. I think it would be nicer for the reader if you could combine the discussion of these paragraphs, by mentioning the relevant general statistics of Table 2 along with the qualitative discussion of Figure 4.**

Similarly, to Comment 7, we believe that this section is well structured to convey the information in a way that is easily understandable by the reader

13. **Page 10, Line 252:** "we choose two special cases".
**If possible indicate how the 2 cases chosen are special in compared to the rest of the dataset.**

We chose these two cases as they contain measurements from clouds of the same system but in different evolutionary stages, which is also our point of interest. The warm conveyor belt case was also outstanding because of the semi-Lagrangian flight path that was followed, allowing us to measure a system along its axis of evolution. The use of the word 'special' might have been misleading and was removed. We have revised this short paragraph as follows:

In order to study the temporal evolution of cirrus clouds and more specifically the changes in RHi through various stages of the cloud's lifetime we choose two cases where a cloud system was measured over an extensive time period and different cloud evolutionary stages were captured.

14. **Figure 6, 8: Please include legents in the plot's colorscales. And values (or mention in the legent if unitless).**

We have added a title to the colorscales informing the reader that the variable that is depicted is the backscatter ratio (BSR). This is also explained in the legend of each figure. Regarding the units, since the plots depict a ratio which by default has no units we believe it is not necessary to add an explanation on that matter.

15. **Figure 7:** "The number of layers differs depending on the depth of the cloud".
**This is also depending on the lidar penetration in the cloud (especially for p2 phase). Consider revising this sentence accordingly.**

Your statement is correct, this is something that should be pointed out. We have revised the legends of Figure 7 and 9 as follows:

The number of layers differs depending on the geometrical depth of the cloud and the depth at which the lidar could provide measurements.

16. **Page 13, Line 308:** "… Gensch et al. (2008), Kübbeler et al. (2011), Petzold et al. (2017), Kaufmann et al. (2018), and Krämer et al. (2009, 2020) also found an RHi mode close to or at ice saturation for mid-latitude cirrus clouds".
**It would be nice to mention the type of dataset used for these studies, i.e. in-situ measurements, radiosondes, lidars?**

The sentence has been revised as follows in order to contain the information on the types of data used by the cited studies.

Ovarlez et al. (2002), Ström et al. (2003), Kübbeler et al. (2011), Petzold et al. (2017) and Kaufmann et al. (2018) using in-situ measurements, Gensch et al. (2008) using in-situ measurements and a model, Comstock et al. (2004) using ground-based raman lidar measurements and Krämer et al. (2009, 2020) using in-situ and satellite remote sensing (lidar and radar) measurements, also found an RHi mode close to or at ice saturation for mid-latitude cirrus clouds.

17. **Page 14, Line 342:** "Voigt et al. (2010) also find the cloud-free RHi values to be mostly subsaturated, but higher than 70 % RHi".
**Please improve the syntaxes of this sentence to make it clearer.**

The phrase has been revised as follows:

Voigt et al. (2010) also report cloud-free RHi mostly below saturation with most of the data between 70% and 100% RHi.

18. **Page 14, Line 366:** "Liquid-origin clouds are thus more frequently measured in environments with ever so slightly lower updrafts, than in-situ clouds".
**One would expect that from the way the liquid-origin clouds were defined in section 2.3 they would be in environments with stronger vertical motions (updrafts and downdrafts in the cloud) than the in-situ clouds, while the sentence is arguing the opposite. If possible, it would be nice if the authors can enhance this discussion (e.g. by adding some other references supporting this statement) or provide a comment on why this may be observed.**

Thank you for your comment. After a detailed review and recalculation we found out that it was actually a mistake. We measure liquid-origin clouds more frequently in slightly higher updrafts. We have changed the text as follows:

…measured clouds. For the in-situ as well as the liquid -origin clouds the most common updraft speeds are very close to 0. Liquid-origin clouds are more frequently measured in slightly higher updrafts, but the difference in frequency is small. Krämer et al., 2016 also expect liquid-origin clouds in higher updrafts but consider clouds from WCB systems to be slow updraft liquid-origin clouds. In our study most liquid-origin clouds are also characterized as stemming from WCBs, supporting the small difference between the two cloud types.

19. **Page 15, line 397:** "The use of a lidar instrument is advantageous over in-situ measurements, as it provides a 2D curtain including the complete vertical structure of the measured clouds. This way one overflight is sufficient enough to measure a whole cloud or cloud system..". **The lidar signal gets often totally attenuated as it measures through a cloud (which is also evident in the 2 case studies of this work). It would be better if the authors rephrase this part to be more accurate to the real capabilities of a lidar instrument.**

    We have revised this phrase as follows to clarify that a lidar is not capable of measuring the whole atmospheric profile under all possible conditions.

    The use of a lidar instrument is advantageous over in-situ measurements, as it provides a 2D curtain of the measured clouds which under favorable conditions can depict even their complete vertical structure. This way one overflight is sufficient enough to measure a whole cloud or cloud system

20. **Page 16, Line 422:** "Lidar measurements of cirrus clouds in high latitudes are scarce and necessary in order to compare their characteristics with mid-latitude cirrus clouds and investigate possible differences".
    **Lidar measurements of cirrus clouds in high latitudes are not scarce, considering the 15-year record of CALIPSO dataset. But the lidar water vapor measurements are. Consider rephrasing this part.**

    Thank you for this comment. We have revised this phrase to better specify our statement as follows:
    Measurements of the water vapor profile of cirrus clouds in high latitudes as provided by airborne lidar instruments are scarce and necessary in order to compare their characteristics with mid-latitude cirrus clouds and investigate possible differences.

**Technical corrections:**
1. Page 1, line 16: "..ML-Cirrus data-set**,** there are two.."      Add comma
2. Page 10, line 250: "..lower than **the ones in** the deeper.." Add to text
3. Page 11, line 273: "..**in** the young cells..."                          Correct as is.
4. Page 11, line 278: "**In** the third part.."                               Add to text
5. Page 14, line 369: "Young, newly formed**,** clouds.."          No comma needed

Technical corrections 1,2 and 4 have been implemented in the text. Suggested corrections 3 and 5 were checked and we believe they are correct in their original form.